# Assessing the Tidal Volume through Wearables: A Scoping Review

**DOI:** 10.3390/s21124124

**Published:** 2021-06-16

**Authors:** Vito Monaco, Cesare Stefanini

**Affiliations:** 1The BioRobotics Institute, Scuola Superiore Sant’Anna, 56127 Pisa, Italy; cesare.stefanini@santannapisa.it; 2Department of Excellence in Robotics & AI, The BioRobotics Institute, Scuola Superiore Sant’Anna, 56127 Pisa, Italy; 3Healthcare Engineering Innovation Center (HEIC), Khalifa University, Abu Dhabi 127788, United Arab Emirates

**Keywords:** tidal volume, wearable sensors, respiratory sensors, physiological monitoring, vital signs

## Abstract

The assessment of respiratory activity based on wearable devices is becoming an area of growing interest due to the wide range of available sensors. Accordingly, this scoping review aims to identify research evidence supporting the use of wearable devices to monitor the tidal volume during both daily activities and clinical settings. A screening of the literature (Pubmed, Scopus, and Web of Science) was carried out in December 2020 to collect studies: i. comparing one or more methodological approaches for the assessment of tidal volume with the outcome of a state-of-the-art measurement device (i.e., spirometry or optoelectronic plethysmography); ii. dealing with technological solutions designed to be exploited in wearable devices. From the initial 1031 documents, only 36 citations met the eligibility criteria. These studies highlighted that the tidal volume can be estimated by using different technologies ranging from IMUs to strain sensors (e.g., resistive, capacitive, inductive, electromagnetic, and optical) or acoustic sensors. Noticeably, the relative volumetric error of these solutions during quasi-static tasks (e.g., resting and sitting) is typically ≥10% but it deteriorates during dynamic motor tasks (e.g., walking). As such, additional efforts are required to improve the performance of these devices and to identify possible applications based on their accuracy and reliability.

## 1. Introduction

The assessment of respiratory activity based on wearable devices is becoming an area of growing interest due to the wide range of available sensors [1,2,3,4,5]. In this framework, the term “wearable device” refers to a cosmetically acceptable accessory that can be donned by the user without encumbering or restricting her/his daily activities and mobility. The possible fields of applications for these devices involve clinical, occupational, and recreational settings, and they are forcing both researchers and companies to develop suitable strategies to monitor the overall status of the user [1,2,3,4,5]. As such, wearable technologies are typically designed to recognize abnormal physiological conditions early; such conditions can reflect chronic respiratory diseases [6], serious adverse events such as cardiac or respiratory arrests [7], sleep apnea [8], or other psycho-physiological stressors [4].

During the last few decades, several wearable devices have been purposely developed to monitor the respiratory functions of healthy subjects and patients in both daily life and clinical settings [1,4,7]. In this respect, many authors mostly focused their attention on the respiratory rate (i.e., number of breaths per minute), as exhaustively documented by recent review studies [1,4]. Indeed, the respiratory rate can accurately diagnose patients at risk of cardio-pulmonary diseases and those requiring admission to intensive care units [9,10], and it is sensitive to other factors such as cognitive load, emotional stress, pain, environmental challenges, discomfort, and rate of physical activity [4].

While useful, the assessment of the respiratory rate is not fully indicative of lung conditions, as it does not provide information pertaining to the actual gas transfer in the lungs. In addition, the respiratory rate is sometimes assessed by analyzing features in the frequency domain [11,12], which may result in inaccurate results in the case of irregular/transient breathing due to unpredictable events (e.g., sleep apnea and cough), sudden exacerbation of the clinical status, or dynamic motor tasks [13,14]. Conversely, the assessment of the tidal volume, i.e., the volume of air that is inhaled and exhaled with each breath [15], is better suited to describe the ventilatory status of a subject [16].

In more detail, the assessment of the tidal volume contributes to the ability to accurately predict cardio-pulmonary complications [17], can allow the monitoring of respiratory issues such as the Cheyne–Stokes respiration and sleep apnea [18,19], is sensitive to physical activity [20,21], and can reflect different respiratory conditions, including obstructions [22,23]. In addition, changes in the tidal volume are ascribed to changes in lung elasticity due to ageing [24] and chronic respiratory diseases [25,26], and it is of paramount importance to identify lung-protective ventilation strategies [27]. Remarkably, a recent study on the COVID-19 pandemic pointed out that incoming patients at the emergency department show variable lung compliance due to both their overall clinical picture and the ongoing spread of the disease [28]. Hence, it is possible to infer that the continuous assessment of the tidal volume, both in clinical settings and at home, is expected to provide helpful indications to properly tune ventilation-based respiratory treatments, or to highlight a rapid decline inrespiratory capabilities in these patients. Finally, the assessment of the tidal volume through noninvasive approaches is among the most convenient strategies to test respiratory functions in subjects, such as children, who lack cooperation when undergoing spirometry [29], and it allows for the estimation of other relevant parameters, such as the inspiratory and expiratory time, which complement the measurement of the pulmonary ventilation.

Given the relevance of the tidal volume, we designed a scoping review aiming to identify research evidence supporting the use of wearable devices to continuously monitor the tidal volume in both daily activities and clinical settings. In more detail, this study was conceived to do the following: (i) highlight the available technology; (ii) clarify the accuracy of the proposed methodological approaches with respect to the experimental conditions (e.g., resting on a bed vs. cycling), as they do affect cardio-respiratory functions [30] and related assessments [31]. In a long-term perspective, this study is expected to contribute to the definition of novel methodological paradigms for an accurate assessment of the tidal volume, which can be used for either the daily monitoring of the users’ status or the development of adaptive respiratory treatments.

## 2. Materials and Methods

### 2.1. Research Question

The methodological approach adopted for this scoping review is reported in the work authored by Arksey and O’Malley [32] and commented on by Colquhoun and colleagues [33]. Specifically, our exploratory research aimed to map technologies and evidence supporting the use of wearable devices for the assessment of the tidal volume, to highlight possible gaps in the literature, and to synthesize existing knowledge in this field.

### 2.2. Search Strategy

A detailed literature search was completed in December 2020 using the following databases: Pubmed, Scopus, and Web of Science. The adopted queries included the following keywords in the title or in the abstract:[respirat* OR breath*]AND[tidal volume]AND[wearable* OR sensor* OR acceler* OR gyroscope* OR inertial OR IMU* OR MIMU* OR Holter OR sound]AND[monitor* OR assess* OR plethysmography]

Only documents in English were retained. No restriction was applied regarding the gender and age of the participants. In the case of multiple copies of the same article, a single copy was retained. Conference abstracts were excluded.

### 2.3. Review Process

The titles and the abstracts of the list of documents previously identified were preliminarily screened to include studies dealing with the topic of this work. A document was retained if it was considered acceptable by at least one of the authors. Then, a full-text evaluation was independently undertaken by the two authors to retain only documents that met the following eligibility criteria:The study compares one or more methodological approaches for the assessment oftidal volume, with an outcome of either state-of-the-art measurement devices dealing with the direct measurement of the amount of air inhaled and exhaled (e.g., spirometer) or the respiratory-related variation in the whole chest-abdomen volume via optoelectronic plethysmography;The approach described in the paper is designed to be exploited in a wearable device, as defined above; accordingly, studies dealing with the analysis of the air flow through masks or similar supports were excluded as these devices stigmatize respiratory disease and, accordingly, might not produce acceptable results by the user during daily activities;The study reports the measurement accuracy; notably, documents showing the estimated tidal volume and not reporting any metrics quantifying the goodness of such values were discarded.

Concerning the measurement accuracy, we can anticipate that all methodological approaches suited for wearable solutions provide an indirect measure of the tidal volume through a calibration process. In other words, they do not directly assess the volume of air inhaled and exhaled with each breath; rather, they assess other physical variables (e.g., changes in chest circumference while breathing) that can be related to the tidal volume through a calibration model, which is typically linear. In this respect, the accuracy of the estimation among the authors is supposed to be assessed in terms of volumetric error, both absolute and relative, and goodness of the fitting model, as represented by the correlation coefficients (ρ) or the coefficient of determination (R^2^). For the purposes of this study, measures representing the qualitative assessment of the accuracy were reported as aggregated across participants. In addition, for studies showing the measurement accuracy in different experimental conditions, we reported the range.

If a document was considered acceptable by only one of the authors, it was discussed to find a consensus about retaining or excluding it. An additional manual search was supplemented by screening the references of these articles to avoid the possibility of overlooked articles.

A study was excluded if it did not deal with the assessment of the tidal volume, involved only animal (not human) models or only platforms simulating breathing mechanics, or did not compare the results with well-established state-of-the-art measurement tools. In addition, studies dealing with the assessment of other features of the tidal volume (e.g., tidal volume variability, flow, and minute ventilation) that did not report information concerning the tidal volume were discarded.

## 3. Results

### 3.1. Literature Search Yield

The electronic screening on merged databases yielded 1031 documents. After merging records among databases, title and abstract screening allowed us to discard 939 documents; thus, we only retained 92 documents. According to the eligibility criteria, 31 documents passed the in-depth screening. Five further documents were added after screening references. Therefore, the final number of retained papers was 36. Figure 1 summarizes the flowchart of the reviewed documents.

### 3.2. Study Characteristics

Table 1 summarizes the main studies characteristics. Overall, we collected 27 journal and 9 conference papers. Typically, the topics of these documents consisted of investigating the feasibility and the accuracy of one or more alternative approaches for the assessment of the tidal volume and, in some cases, other respiratory variables, under different experimental conditions.

Based on the summary reported in Table 2, the sensing elements across the studies were the following: optical sensors, resistive/inductive stretch sensors, accelerometers, tactile/pressure sensors, approaches dealing with the analysis of the electrocardiograms (ECGs) or the electrical impedance, electromagnetic sensors, and acoustic sensors. In 13 of the 36 documents, the authors analyzed the metrological features of four different commercial devices: Airgo^TM^ (MY AIR Inc., Boston, MA, USA; MyAirGo Italy Srl, Milan, Italy); Lifeshirt^TM^ (VivoMetrics, Ventura, CA, USA); Respitrace^®^ bands (Ambulatory Monitoring, Inc., White Plains, NY, USA); NOX T3 Sleep Monitor (NoxMedical, Reykjavík, Iceland). Noticeably, the sensing principle of commercially available devices was based on either inductive plethysmography or resistive stretch sensors. The remaining 23 documents introduced different strategies to assess the tidal volume based on alternative measurement strategies.

### 3.3. Participants and Experimental Protocol

Table 3 summarizes the enrolled participants and experimental protocols across reviewed studies. Most of them (i.e., 31 out of 36) involved only healthy subjects with ages in the range of 18–61 years old and a wide range of anthropometrical characteristics. The enrolled participants were both female and male in 14 studies and only male or only female in 8 and 1 studies, respectively. In the remaining documents, i.e., 8 out of 31, the authors did not report the gender of the enrolled participants. A limited set of studies involved either both healthy subjects and patients (3 out of 36 studies) or only patients (2 out of 36 studies). Noticeably, patients were usually adults (age ≥49 years old) or children with ages in the range of 3–8 years old. Two of these 5 studies involved both female and male participants; 2 of these 5 studies involved one participant only. The number of enrolled participants across all 36 studies typically ranged between 10 and 20 persons even if its distribution was skewed toward lower values. For one paper, the number of enrolled participants was significantly higher, i.e., 186 persons [48].

As far as the experimental protocol was concerned, 23 of the 36 documents investigated the correspondence between the outcome of a reference measurement device (e.g., spirometer) with that of a wearable device during quasi-static activities, such as sitting, standing, laying on a bed, reclining, meditating, and sleeping. In many of these studies, participants were asked to perform standardized respiratory maneuvers (e.g., variable tidal volume, paced breathing, and emulating shallow, apnea, obstruction, and other pathological breathing patterns), with or without visual feedback. Additionally, changes in posture (e.g., dorsal decubitus, left lateral decubitus, right lateral decubitus) or the movements of different body segments (e.g., head, arms, and legs) were also accounted for during the experimental sessions. In 13 of the 36 studies, participants were asked to undertake either both quasi-static and dynamic tasks or dynamic tasks only. In this respect, dynamic tasks consisted of cycling, walking, and running at different speeds, and performing manual housework tasks.

### 3.4. Assessment of the Accuracy

As reported in Section 2.3, the assessment accuracy was usually evaluated in terms of the volumetric error and goodness of the fitting model between tidal volumes obtained by the proposed approach and those resulting from a reference measurement system. However, metrics adopted to quantify the volumetric error in particular were different among the authors. Table 4 provides an overview of the values reported across all studies.

In studies dealing with quasi-static motor tasks, the relative volumetric error averaged across the subjects, expressed as a percentage of the volume measured by the reference measurement tool (see RelErr in Table 4), was typically lower than 20%. Noticeably, in some cases, the RelErr fell below 5% [31,38,58], whereas, in one case, values appeared to be significantly higher (+35%; [57]). When the volumetric error was assessed as the difference between estimated and reference values (see MOD in Table 4), the authors typically reported an averaged bias across enrolled participants lower than 0.1 L. Very often, the bias was one order of magnitude lower, whereas, in some cases, it had significantly higher values (e.g., from 0.7 to 5 L; [43]). Concerning the goodness of the fitting model, both the correlation coefficients (see ρ in Table 4) and coefficients of determination (see R^2^ in Table 4) were usually greater than 0.75, with a distribution skewed toward higher values. For a limited set of studies only, fitting models were characterized by a lower goodness of fit [19,63].

Studies dealing with dynamic tasks reported values for the relative volumetric error averaged across participants lower than 40%, or a bias comparable to values reported for quasi-static motor tasks (MOD < 0.1 L). Correlation coefficients and coefficients of determination were comparable to those reported for quasi-static motor tasks (ρ and R^2^ were usually greater than 0.85).

## 4. Discussion

This scoping review aimed to identify research evidence supporting the use of wearable devices to continuously monitor the tidal volume in both daily activities and clinical settings. As such, it is expected to contribute to the definition of novel methodological paradigms for an accurate assessment of the tidal volume based on wearables, which can be used for either the daily monitoring of the users’ status or the development of adaptive respiratory treatments.

The results revealed that several approaches have been investigated in the literature, mainly involving stretch sensors with different sensing principles (i.e., resistive, inductive, optical, and pressure/tactile; see Table 2). For these devices, the tidal volume is indirectly assessed as the variation inthe chest and abdomen circumference through a calibration model. Other approaches, based on inertial measurement units (IMUs), acoustic sensors, or the assessment of suitable electrophysiological signals from the trunk have also been explored; however, their reported results appear to be quite preliminary (Table 2 and Table 3). As expected, the accuracy of the assessment was higher in studies dealing with quasi-static motor tasks, which reported a relative error percentage in the neighborhood of 10%, likely corresponding to about 0.05–0.10 L (Table 4). During dynamic motor tasks, the accuracy in terms of the relative volumetric error typically decreased despite the fact that the difference between the estimated and reference volumes was comparable to values observed under quasi-static conditions (Table 4).

Despite the wide range of proposed solutions, some of which are commercially available (Table 2), we observed that studies aiming to broadly test their performance among different conditions (e.g., clinical setting vs. activities of daily living, static vs. dynamic motor tasks) and potential users (e.g., patients vs. healthy subjects, young vs. old individuals, male vs. female) are marginal. Specifically, experimental tests have typically been performed thanks to the involvement of 10–20 or fewer individuals, and only in a narrow set of documents did the authors purposely account for subjects with different anthropometrical features or gender-based dimorphisms (Table 3). Finally, the absence of a standardized strategy among studies to quantify the accuracy (Table 4), as well as the limited number of papers explicitly reporting failures in their proposed approaches, suggests that the assessment of the tidal volume via wearable sensors deserves additional investigation. In this respect, according to previous authors [65], further studies should be designed to identify standardized methodological procedures to both quantify the accuracy among different approaches and estimate the risk of failure (e.g., false alarms) of these devices in order to prevent dangerous consequences for the users.

### 4.1. Sensor-Based Taxonomy

Despite the wide range of the possible sensing principles used to assess the tidal volume (Table 2), we observed that retained documents can be roughly classified into two groups: i. the first group accounts for all papers dealing with optical, resistive, and inductive stretch sensors; ii. the second group accounts for the remaining strategies, based on IMUs, pressure and tactile sensors, sensing algorithms parsing electrocardiogram or electrical impendence, and acoustic and electromagnetic sensors. For the former group, at least six documents for each sensing principle were reviewed; for the latter, we found between one and three documents, suggesting that these strategies are still at an early stage. Based on this evidence, for the following sensory-based taxonomy, we pooled together all documents referring to the latter group.

#### 4.1.1. Optical Sensors

Nine of the retained studies tested the feasibility and accuracy of using an optical sensor-based approach to assess the tidal volume (Table 2) [12,34,50,52,54,55,56,58,60]. Noticeably, these studies have been carried out by five independent research groups; thus, some of the reported results appeared redundant.

Two main sensing principles have been proposed among the studies. The first consists of an optical structure inducing a change in the spectral profile of the light traveling along the sensor due to bending (long-period grating sensors [34,50,58,60]; few-mode fiber Mach–Zehnder interferometer [12]) or strain (fiber Bragg grating sensors [54,55,56]). These sensors are usually embedded (glued or stitched) in fitting garments or suitable elastic bands, and they are designed to monitor changes in the chest and abdomen curvature/circumference while breathing. Remarkably, this approach is intrinsically magnetic-resonance compatible, as highlighted by Massaroni and colleagues [55]. The second approach is based on optical encoders sliding over an inextensible code strip coupled with stretchable bands; it is sensitive to changes in trunk circumference due to breathing [52]. In both cases, sensors can be arranged in a network whose output is parsed out by a suitable calibration model to provide an estimation of the tidal volume.

#### 4.1.2. Resistive Stretch Sensors

Six reviewed studies dealt with the assessment of the tidal volume measured by resistive stretch sensors (Table 2) [6,31,35,45,46,51]. Two of these studies [31,35] tested the accuracy of a commercial device, namely Airgo^TM^, while estimating the tidal volume in quasi-static (e.g., standing, sitting, and supine) and dynamic (i.e., cycling) motor tasks (Table 3). Airgo^TM^ is a standalone sensory platform consisting of an extensible band designed to provide a signal proportional to the thoracic circumference. It is also equipped with a triaxial accelerometer to monitor the user’s daily activities, and it can be connected to a network infrastructure for telemedicine. Other authors investigated the performance of a set of electro-resistive bands arranged in suitable garments, such as a t-shirt [45,46] or pants [51]. These approaches rely on more complex measurement systems compared to a single standalone band. However, due to the limited number of enrolled participants (1 subject for [45,46], 4 subjects for [51]) and the quasi-static experimental conditions, the reported results appear to be preliminary. A recent study introduced disposable stretch sensors composed of a piezoresistive metal thin film set in a silicone elastomer substrate with a very small footprint (21 mm × 10 mm × 0.5 mm) [6]. A couple of these sensors have been attached to the skin of healthy subjects to assess the tidal volume while performing both quasi-static and dynamic (walking and running) motor tasks (Table 3), and they showed good accuracy at least during the quasi-static experimental tests (Table 4).

#### 4.1.3. Inductive Stretch Sensors

Eleven documents investigated the accuracy of inductive stretch sensors while assessing the tidal volume (Table 2) [36,38,40,41,42,47,48,49,61,63,64]. The sensing principle underlying this approach consists ofmeasuring changes in the self-inductance of two sinusoidal coils surrounding the abdomen and/or rib cage due to respiratory movements. This approach, also called respiratory inductance plethysmography (RIP), is purported to be among the most suitable methods for the assessment of ventilation. As such, it has been widely tested in several studies in both quasi-static and dynamic conditions (Table 3). Noticeably, all the reviewed studies dealing with RIP measured the accuracy of commercial devices, that is, LifeShirt^TM^ (VivoMetrics, Ventura, California, USA), a multi-channel ambulatory device embedding insulated sinusoidal wires surrounding the rib cage and abdomen, Respitrace^®^ (Ambulatory Monitoring, Inc., New York) inductance bands, and a polysomnography system (NOX T3 Sleep Monitor, NoxMedical, Reykjavík, Iceland) allowing the measurement of nasal pressure and thoracic RIP.

All of the reviewed studies were based on evidence that the RIP is accurate enough when detecting respiratory variables in clinical/ambulatory settings, that is, in almost resting conditions. Accordingly, the authors explored the extent to which RIP can be accurately used with unrestrained subjects. Specifically, the authors of these studies investigated suitable strategies to automatically calibrate the sensor based on position sensory feedback mediated by an IMU [36], to assess the accuracy of RIP during or across dynamic motor tasks [38,40,47,48,61,63,64], to assess the accuracy of RIP while coupled with technical garments [41], and to determine the between-day variability of calibration parameters [49]. Eight of the ten studies enrolled healthy young adults [38,41,47,48,49,61,63,64], while the remaining two enrolled patients with chronic obstructive pulmonary disease and congestive heart failure [36,40].

#### 4.1.4. Alternative Strategies

Two of the retained papers dealt with the use of a triaxial accelerometer located on the chest to estimate the tidal volume based on the reconstruction of the angular motion induced by breathing (Table 2) [13,44]. In both cases, the accelerometer was used to assess the orientation of the chest wall while breathing by assessing the misalignment between one of its axes and the gravity. Bates and colleagues enrolled both healthy subjects, mostly for validating the sensor calibration, and post-operative patients for an overnight capture session [13]. Fekr and colleagues enrolled healthy subjects who were asked to breathe according to different patterns (e.g., Normal, Bradapnea, Tachypnea, Kussmaul, and Cheyne–Stokes) [44], and they reported that the breathing pattern was more variable, while the correlation coefficient was less variable.

Three documents investigated the accuracy of either pressure [19,37] or tactile sensors [43]. Caldiroli and Minati explored the feasibility of using remote pressure sensor plethysmography to monitor children undergoing sedation in an MR scanner [37]. However, the authors could only report the outcome of the proposed sensory strategy compared to that assessed by a high-precision spirometer in healthy subjects underquasi-static conditions. Hoffman and colleagues evaluated the accuracy of capacitive textile pressure sensors embedded in a band wrapped around the chest compared to that of a pneumotachograph in healthy subjects during sleeping, walking, and running [19]. Earthrowl-Gould and colleagues reported preliminary results related to an inexpensive tactile chest sensor, consisting of a sandwich of two stiff plates of copper-coated board and conductive foam packaging; its resistance changed with the foam thickness due to respiratory movements [43].

Other authors explored the hypothesis that electrophysiological signals, such as the electrocardiogram [53] and the electrical impedance [42,62], can provide an indirect measure of the tidal volume. Lázaro and colleagues proposed a pilot study aiming to estimate changes in the tidal volume by analyzing standard ECG signals underquasi-static conditions. Cohen and colleagues [42] compared impedance and inductance ventilation sensors and found no significant difference between the two approaches. Seppä and colleagues confirmed the accuracy of the assessment of the tidal volume via impedance pneumography in children with wheezing disorders [62].

Two preliminary studies investigated the accuracy of acoustic sensors for a noninvasive measurement of ventilation under quasi-static conditions [39,59]. For both studies, a microphone was hosted in a small case and placed over the trachea. Despite the fact that the reported accuracy was comparable to other measurement systems, the authors pinpointed potential limitations due to the effects of the background noise on the outcome of the sensors.

Lastly, Padasdao and colleagues developed an electromagnetic transducer converting the variations in chest circumference into the rotation of a small DC brushed motor used as a voltage generator [57]. This approach was tested for both static motor tasks (i.e., sitting and standing) and walking at different speeds. However, the comparison between the outcome of the proposed sensor and that of a spirometer provided contrasting results. Indeed, despite the fact that the reported mean absolute error between the two measurement systems was greater than 30%, the mean of their difference was comparable with other, more accurate approaches across all experimental conditions (i.e., in the worst case −0.021 ± 0.225 mL; see Table 4). Interestingly, the proposed approach is capable of harvesting respiratory energy; thus, it can potentially reduce the overall power requirement.

### 4.2. Comparative Works

Some review studies focused their attention on topics somewhat related to our one. More in detail, Folke and colleagues critically reviewed noninvasive methods and devices that have been claimed to provide information about the respiratory rate or depth, or gas exchange for clinical settings [65]. Dinh and colleagues reviewed the recent development of stretchable physical sensors for monitoring respiration signals [1]. Other authors aimed to review current wearable strategies to assess vital signs [3,5]. However, to the best of our knowledge, our study is the only one that is purposely focused on the assessment of the tidal volume based only on wearable sensors. Accordingly, we believe that it extends the previous findings and can contribute to identifying the most promising methodological paradigms for the assessment of the tidal volume.

## 5. Conclusions

The key component of wearable devices for the assessment of the tidal volume, that is the sensor, can be realized mainly by using different technological solutions, ranging from IMUs to strain sensors (e.g., resistive, capacitive, inductive, electromagnetic, and optical) or acoustic sensors. They, respectively, exploit the movements of the chest wall while breathing, or the sound generated by the air flow through respiratory paths, as an indirect measure of respiratory activity. In spite of this apparently wide array of possibilities, it should be acknowledged that the continuous monitoring of the tidal volume is extremely challenging. Breathing patterns can be characterized by a significant inter-cycle variability, which can be further altered by pathologies, behaviors, or unpredictable events such as apnea or cough. In addition, the outcome of methodological approaches dealing with the assessment of respiratory-related chest wall mechanics can be affected by artifacts such as body movements or the incorrect positioning of sensors.

The results of our review study highlighted that some of the proposed methodological approaches (i.e., optical, resistive, and inductive stretch sensors) have been widely investigated in the literature, while the others are at an early stage. In both cases, our analysis revealed very interesting solutions that can potentially boost the development of such devices (e.g., the disposal resistive stretch sensors described in [6]; the electromagnetic device for sensing and energy harvesting introduced in [57]). However, the accuracy of the proposed wearable solutions might not be satisfying enough. As a matter of fact, current recommendations concerning calibration checks in spirometry suggest that the maximum relative error should fall within ±3.5% to prevent significant inaccuracies [66]. Wearable sensors can instead only guarantee a relative volumetric error underquasi-static conditions equal to 10% or greater, which might not be acceptable for some monitoring situations (e.g., clinical settings). When subjects are involved in dynamic tasks (e.g., walking, running, and doing homework), the error increases further. Accordingly, despite the wide range of available solutions, in terms of taxonomy and configurations, future research should be oriented to mainly improve the performance of the wearable devices and to identify possible applications based on their accuracy and reliability.

## Figures and Tables

**Figure 1 sensors-21-04124-f001:**
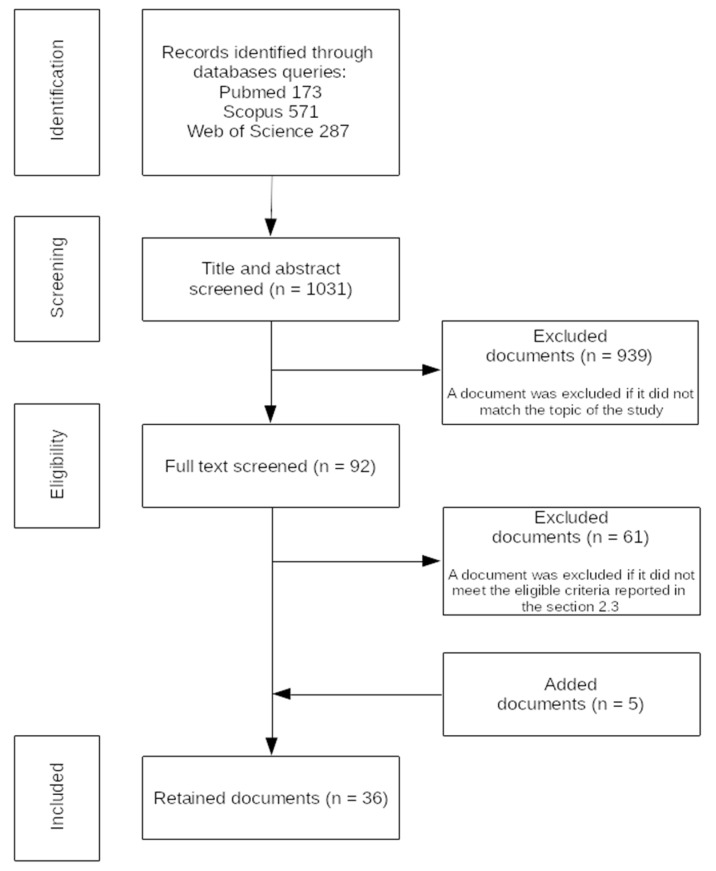
Flow chart of the article selection process.

**Table 1 sensors-21-04124-t001:** Description of reviewed studies.

Reference	Type ^1^	Aim of the Study
Allsop et al., 2007 [34]	JP	To investigate the accuracy of an array of long-period grating sensors interrogated by a set of distributed feedback lasers to assess the curvature of chest and abdominal regions.
Angelucci et al., 2020 [35]	JP	To test the performance of a 5G-based telemonitoring system accounting for three commercial monitoring devices: **i.** the Airgo^TM^ (MYAIR Inc., Boston, Massachusetts; MyAirGo Italy Srl, Milan, Italy); **ii.** the uHoo environmental sensor (uHoo, Kwun Tong, Hong Kong); **iii.** the SAT-30 finger pulse oximeter (ContecNedical Systems CO, Ltd., Qinhuangdao, China).
Antonelli et al., 2020 [31]	JP	To compare a metabolic cart with Airgo^TM^ (MYAIR Inc., Boston, Massachusetts; MyAirGo Italy Srl, Milan, Italy) to derive breathing parameters from body surface motion detection acquired at the level of the lower ribcage.
Bates et al., 2010 [13]	CP	This paper presents a novel algorithm for obtaining a respiration signal of the patient at rest from triaxial accelerometer data, which is validated against the flow rate waveforms as measured by nasal cannula pressure transducers.
Brullman et al., 2010 [36]	JP	To develop a method for position-specific automatic calibration of respiratory inductance plethysmography (RIP) during changes in body position based on position sensor feedback.
Caldiroli et al., 2007 [37]	JP	To explore the technical feasibility of breathing pattern monitoring based on remote pressure sensor respiratory plethysmography in the MR scanner, and its viability for use in the clinical routine.
Caretti et al., 1994 [38]	JP	To compare the accuracy of RIP and a simultaneous flowmeter of tidal volume during rest, graded treadmill, and cycling exercises.
Chen et al., 2014 [39]	CP	To show the initial results of a simple method to automatically estimate lung tidal volumes using the outputs from a very small wireless acoustic-based sensor placed on the suprasternal notch.
Chu et al., 2019 [6]	JP	To measure both respiration rate and volume using a disposable wearable strain sensor placed discreetly on the abdomen and ribcage.
Clarenbach et al., 2005 [40]	JP	To evaluate the accuracy of a portable commercial respiratory inductive plethysmograph (Lifeshirt, VivoMetrics, Ventura, Calif.) that allows the monitoring of ventilation without airway instrumentation during exercise in unrestrained subjects.
Coca et al., 2010 [41]	JP	To establish that a commercial sensor vest, i.e., Lifeshirt (VivoMetrics, Ventura, Calif.) could provide physiological information, especially those pertaining to respiratory functioning, comparable to standard physiological equipment, and could continue to provide these data to remote receivers while being exposed to challenging micro-environmental conditions (skin-related temperature and humidity).
Cohen et al., 1997 [42]	JP	To compare the effects of motion and airway obstruction on impedance plethysmography (IP) and respiratory inductive plethysmography (RIP).
Earthrowl-Could et al., 2001 [43]	JP	To investigate the feasibility of an array of tactile sensors that react to local chest and abdominal movements and provide signals that can be related algorithmically to those generated by conventional spirometric devices.
Fekr et al., 2014 [44]	CP	To develop a cloud-based tool for accurately monitoring the respiration patterns of patients with a three-axis accelerometer sensor.
Gargiulo et al., 2015a [45]	JP	To investigate the use of commercially available electro-resistive bands (ERBs) to achieve a simultaneous measurement of cardiac stroke, lung monitoring, and respiration effort.
Gargiulo et al., 2015b [46]	JP	To investigate the use of commercially available electro-resistive bands (ERBs) to achieve a simultaneous measurement of cardiac stroke, lung monitoring, and respiration effort during sleeping.
Grossman et al., 2010 [47]	JP	To compare the assessment of ventilator parameters, based on ambulatory inductive plethysmography and a mobile backpack flowmeter-derived ergospirometry system employing a closed facemask (Oxycom Mobile, Jaeger), across different body posture and with/without mental efforts.
Heyde et al., 2014 [48]	JP	To examine the validity of posteriori-adjusted gain factors and the accuracy in resultant breath-by-breath respiratory inductance plethysmography data recorded under exercise conditions to provide a rationale (‘‘best possible fit’’) for future validations of a priori calibration procedures.
Heyde et al., 2015 [49]	JP	To examine the between-days variability in determined gain factors (via least square regression) for wearable respiratory inductance plethysmography and its validity for the repeated use in subsequent measurements.
Hoffman et al., 2011 [19]	CP	To introduce noninvasive respiration measurements based on textile pressure sensors.
Ivanovic et al., 2016 [50]	JP	To investigate: **i.** the possibility to use a long-period fiber grating curvature sensor to monitor respiration; **ii.** the possibility to use outward/inward torso movement as a more confident trigger compared to the commonly used pneumatic trigger.
Kim et al., 2007 [51]	CP	To develop a wireless system transmitting the abdominal dimension change signals induced by respiration to estimate the tidal volume.
Laufer et al., 2021 [52]	CP	To introduce and evaluate a novel wearable device that can continuously measure tidal volumes via changes in the circumference of the upper body.
Làzaro et al., 2020 [53]	CP	To present how the changes in tidal volume can be tracked by an armband electrocardiogram (ECG)-based wearable device.
Lo Presti et al., 2018 [54]	CP	To test a man fit-based smart textile thatembeds 12 fiber Bragg grating sensors (FBGs) on women for monitoring breath-by-breath temporal respiratory parameters (i.e., respiratory rate, and inspiratory and expiratory phases duration) and respiratory volume collected with the proposed garment against a reference instrument.
Massaroni et al., 2016 [55]	JP	To test the feasibility assessment of a smart textile based on 6 fiber Bragg grating sensors (FBGs) for the monitoring of compartmental volume changes, global volume changes, and breathing rate. This sensor device was also used during magnetic resonance examinations to investigate hazards or artifacts to the images.
Massaroni et al., 2018 [56]	JP	To investigate the feasibility of twelve fiber Bragg grating sensors embedded in smart textiles for (i) the breath-by-breath monitoring of several temporal and volumetric variables.
Padasdao et al., 2018 [57]	JP	To assess tidal volume detection by using an electromagnetic generator as a biosensor; noticeably, the proposed biosensor is expected to harvest respiratory energy to supplement power requirements.
Petrovic et al., 2014 [58]	JP	To investigate the feasibility of a novel apparatus for the continuous monitoring of the tidal volume based on a single long-period grating curvature sensor.
Que et al., 2002 [59]	JP	To estimate airflow from breath sound intensity during quiet breathing and, therefore, almost exclusively on flows of <0.5 L/s.
Raicevic et al., 2015 [60]	CP	To investigate the feasibility of a long-period grating sensor of bending in monitoring respiratory volumes.
Retory et al., 2016 [61]	JP	To develop and validate a method to facilitate the use of RIP during mild physical activities.
Seppa et al., 2013 [62]	JP	Twofold objective: **i.** to investigate the agreement between impedance pneumography and a direct mouth pneumotachograph in tidal flow measurement in preschool children; **ii.** to compare changes in tidal flow characteristics during methacholine-induced bronchoconstriction with mechanical impedance measurement of the respiratory system.
Wang et al., 2020 [12]	JP	To propose a wearable and compact device for respiration rate and tidal volume monitoring using an in-line few-mode fiber Mach–Zehnder interferometer.
Whyte et al., 1991 [63]	JP	To assess the accuracy of the inductive plethysmograph during a night’s sleep in unrestrained normal subjects
Witt et al., 2006 [64]	JP	To investigate the validity of a novel portable respiratory inductive plethysmograph to the change in respiratory parameters during exercise by comparison witha reference standard pneumotachograph.

**^1^ CP**: Conference Paper; **JP**: Journal Paper.

**Table 2 sensors-21-04124-t002:** Main sensing principles.

Reference	Sensing Principle ^1^										Was the Assessment Device Commercially Available?				
	OPT	RSS	IMU	ISS	PS	AS	EI	TS	ECG	EMAG	Airgo^TM^	Lifeshirt^TM^	Respitrace^®^	NOX T3 Sleep Monitor	Not, It Was Not.
Allsop et al., 2007 [34]	■														▲
Angelucci et al., 2020 [35]		■									▲				
Antonelli et al., 2020 [31]		■									▲				
Bates et al., 2010 [13]			■												▲
Brullman et al., 2010 [36]				■								▲			
Caldiroli et al., 2007 [37]					■										▲
Caretti et al., 1994 [38]				■									▲		
Chen et al., 2014 [39]						■									▲
Chu et al., 2019 [6]		■													▲
Clarenbach et al., 2005 [40]				■								▲			
Coca et al., 2010 [41]				■								▲			
Cohen et al., 1997 [42]				■			■						▲		
Earthrowl-Could et al., 2001 [43]								■							▲
Fekr et al., 2014 [44]			■												▲
Gargiulo et al., 2015a [45]		■													▲
Gargiulo et al., 2015b [46]		■													▲
Grossman et al., 2010 [47]				■								▲			
Heyde et al., 2014 [48]				■								▲			
Heyde et al., 2015 [49]				■								▲			
Hoffman et al., 2011 [19]					■										▲
Ivanovic et al., 2016 [50]	■														▲
Kim et al., 2007 [51]		■													▲
Laufer et al., 2021 [52]	■														▲
Làzaro et al., 2020 [53]									■						▲
Lo Presti et al., 2018 [54]	■														▲
Massaroni et al., 2016 [55]	■														▲
Massaroni et al., 2018 [56]	■														▲
Padasdao et al., 2018 [57]										■					▲
Petrovic et al., 2014 [58]	■														▲
Que et al., 2002 [59]						■									▲
Raicevic et al., 2015 [60]	■														▲
Retory et al., 2016 [61]				■										▲	
Seppa et al., 2013 [62]							■								▲
Wang et al., 2020 [12]	■														▲
Whyte et al., 1991 [63]				■									▲		
Witt et al., 2006 [64]				■								▲			
**Tot. number of documents**	**9**	**6**	**2**	**11**	**2**	**2**	**2**	**1**	**1**	**1**	**2**	**7**	**3**	**1**	**23**

^1^**AS**: Acoustic Sensor; **ECG**: Electrocardiogram-based sensor; **EI**: Electrical Impedance; **EMAG**: Electromagnetic Sensor; **IMU**: Inertial Measurement Unit; **ISS**: Inductive Stretch Sensor; **OPT**: Optical; **PS**: Pressure Sensor; **RSS**: Resistive Stretch Sensor; **TS**: Tactile Sensor.

**Table 3 sensors-21-04124-t003:** Enrolled participants and experimental protocol.

Reference	Enrolled Participants ^1^		Experimental Protocol	
	Healthy Subjects	Patients	Quasi-Static Motor Tasks	Dynamic Motor Tasks
Allsop et al., 2007 [34]	6 M Age (year): 41.7 ± 13.6 Mass (kg): 82.9 ± 16.6 Height (cm): 179 ± 012		Participants were monitored while breathing naturally and deeply while standing; 5 records, one minute long, were collected for each subject.	
Angelucci et al., 2020 [35]	5 F + 5M Age (year): N/A Mass (kg): N/A Height (cm): N/A		Participants were monitored while performing a slow vital capacity maneuver at different tidal volumes (from 3 to 0.5 L, step 0.5 L) with the help of visual feedback.	
Antonelli et al., 2020 [31]	11 F + 10 M Age (year): 41.7 ± N/A (24–51) Mass (kg): 67 ± 16 Height (cm): 170 ± 8		Participants were monitored while breathing quietly in five standardized positions (standing, seated, supine, and right and left lateral decubitus).	Participants were monitored while performing incremental exercises consisting ofcycling on a cycle ergometer along a standardized protocol.
Bates et al., 2010 [13]		One post-operative patient	Participants were monitored along a 6.9 h overnight capture.	
Brullman et al., 2010 [36]	5 subjects Age (year): (23–26)	10 patients with COPD Age (year): 49 ± 21 2 patients with RLD Age (year): (49–57)	Participants were monitored while resting supine and standing, and sitting. The experimental session lasted between 60 and 90 min.	Patients were monitored while walking for approximately 10 min.
Caldiroli et al., 2007 [37]	8 subjects Age (year): 25 ± 7		Participants were monitored while lying in the supine position.	
Caretti et al., 1994 [38]	2 F + 6 M Age (year): 22 ± 3 Mass (kg): 73.7 ± 13.0 Height (cm): 175.9 ± 8.0		Participants were monitored during the resting period.	Participants were monitored while performing progressive treadmill and cycling exercises.
Chen et al., 2014 [39]	4 subjects		Participants were monitored while sitting along a 5–10 min-long time window.	
Chu et al., 2019 [6]	3 F + 5 M Age (year): N/A Mass (kg): 65.6 ± 10.9 Height (cm): 172 ± 8		Participants were monitored while in the reclined position during several respiratory maneuvers: - To pace their breathing with an audio-visual metronome (breath rate 10, 20, and 40 bpm for 2, 1, and 0.5 min-long record, respectively); - To take a series of shallow, medium, and deep breaths in any order at their discretion; - To inhale maximally and forcefully exhale, sustaining the exhalation for 6 s.	A participant was monitored during walking and running at different speeds, while carrying out several respiratory maneuvers: - To pace their breathing with an audio-visual metronome (breath rate 10, 20, and 40 bpm for 2, 1, and 0.5 min-long record, respectively); - To take a series of shallow, medium, and deep breaths in any order at their discretion; - To inhale maximally and forcefully exhale, sustaining the exhalation for 6 s.
Clarenbach et al., 2005 [40]	10 Healthy subjects without backpack 2 F + 8 M Age (year): 34 ± 7 BMI [kg/m^2^]: 23 ± 1 10 Healthy subjects with backpack 5 F + 5 M Age (year): 30 ± 7 BMI [kg/m^2^]: 22 ± 2	5 male patients with CHF Age (year): 57 ± 22 BMI [kg/m^2^]: 23 ± 3 5M + 1F patients with COPD Age (year): 62 ± 8 BMI [kg/m^2^]: 25 ± 5		Participants were monitored while performing progressive treadmill exercises to exhaustion with and without a backpack.
Coca et al., 2010 [41]	2 F + 8 M Age (year): (21–39)			Participants were monitored while performing progressive treadmill exercises along a 20 min-long session.
Cohen et al., 1997 [42]	10 M Age (year): (22–61)		Participants were monitored while keeping the supine position, carrying out several respiratory maneuvers (i.e., natural breathing, no breathing, simulated airway obstruction, shallow breathing, deep breathing, simulated yawning, simulated snoring, simulated central apnea, and simulated coughing) in conjunction with arm or leg movements. The record lasted approximately 15 min.	
Earthrowl-Could et al., 2001 [43]	1 Male		Participants were monitored while performing four breaths with variable volume while standing.	
Fekr et al., 2014 [44]	4 F + 4M Age (year): (18–46)		Participants were monitored while performing Normal, Bradapnea, Tachypnea, and Kussmaul patterns, each for 1 min, Cheyn–Stokes and Biot’s breathing exercises each for 2 min, and finally a pattern with different tidal volumes lasting for about 3 min.Experimental trials lasted for about 45 min.	
Gargiulo et al., 2015a [45]	1 subject		Participants were monitored while sitting along a minutetime window.	
Gargiulo et al., 2015b [46]	1 subject		Participants were monitored while lying on a standard household bed after several hours from the calibration.	
Grossman et al., 2010 [47]	5 F + 4 M Age (year): 37.3 ± 10.2 BMI [kg/m^2^]: 22.3 ± 1.98			Participants were monitored while performing several behavioral activities characteristic of everyday life (e.g., sitting, walking at different speeds, manual housework tasks, and talking).
Heyde et al., 2014 [48]	88 F + 98 M Age (year): 27.1 ± 8.3 Mass (kg): 68.9 ± 11.1 Height (cm): 175.6 ± 9.0		Participants were monitored while standing still.	Participants were monitored while performing progressive treadmill running, and recovery period.
Heyde et al., 2015 [49]	5 F + 5 M Age (year): 29.4 ± 9.8 Mass (kg): 65.8 ± 12.0 Height (cm): 174.7 ± 10.4		Participants were monitored while standing still, repeated on 5 different days.	Participants were monitored while performing slow running, fast running, and recovery period, repeated on 5 different days.
Hoffman et al., 2011 [19]	7 F + 11 M Age (year): 27 ± 3 BMI [kg/m^2^]: (18–25)		Participants were monitored while performing several respiratory maneuvers and while sleeping.	Participants were monitored while walking and running.
Ivanovic et al., 2016 [50]	9 F + 9 M Age (year): 43 ± 8 BMI [kg/m^2^]: 24 ± 5		Participants were monitored while performing 1 min of natural and 1 min of shallow breathing in the supine position.	
Kim et al., 2007 [51]	4 M		Participants were monitored while breathing under resting state, additionally taking a maximal inspiration and a few artificial coughs.	
Laufer et al., 2021 [52]	2 F + 3 M Age (year): 28 ± 6 Mass (kg): 71.4 ± 8.6 Height (cm): 176 ± 6		Participants were monitored while varying tidal volume (normal, medium, and maximum).	
Làzaro et al., 2020 [53]	5 subjects		Participants were monitored while varying tidal volume.	
Lo Presti et al., 2018 [54]	8 F Age (year): 22 ± 2 Mass (kg): 58 ± 5 Height (cm): 166 ± 5		Participants were monitored while quietly breathing assuming a standing posture. Two trials, 60 s long each, were collected.	
Massaroni et al., 2016 [55]	6 M Age (year): 26 ± 3 Chest circumf. (cm): 97.2 ± 4.1		Participants were monitored while breathing quietly for a 60 s long trial in a standing posture.	
Massaroni et al., 2018 [56]	8 M Age (year): 24 ± 2 Mass (kg): 69 ± 7 Height (cm): 173 ± 5		Participants were monitored while breathing quietly for a 60 s long trial in a standing posture.	
Padasdao et al., 2018 [57]	20 subjects		Participants were monitored while breathing quietly during sitting down and standing up for two 5 min-long trials.	Participants were monitored while walking at different speeds (each trial lasted 5 min).
Petrovic et al., 2014 [58]	9 F + 9 M Age (year): 43 ± 8 BMI [kg/m^2^]: 24 ± 5		Participants were monitored while performing 1 min of natural and 1 min of shallow breathing in the supine position.	
Que et al., 2002 [59]	7 nonsmoking normal volunteers	1 asymptomatic asthmatic subject	Participants were monitored while quietly breathing in different postures (sitting, standing, and supine) and while carrying out several head movements.	
Raicevic et al., 2015 [60]	15 subjects Age (year): 32.6 ± 8.0		Participants were monitored while performing one minute of natural, shallow, and a combination of deep, natural, and shallow breathing in supine position.	
Retory et al., 2016 [61]	23 F + 7 M Age (year): 36.8 ± 14.3			Participants were monitored while performing several tasks (resting, marching on the spot, knee-rising with swinging arms at different cadences).
Seppa et al., 2013 [62]		5F + 16M affected by lower respiratory tract symptoms Age (year): [3.6–7.9] Mass (kg): [15.5–31.8] Height (cm): [99–125]	Participants were monitored while breathing quietly in a sitting position.	
Wang et al., 2020 [12]	5 subjects Age (year): (23–25)		Participants were monitored while breathing normally, or notbreathing for about 15 s.	
Whyte et al., 1991 [63]	8 M Age (year): (23–33)		Participants were monitored while sleeping in different positions and at each sleep stage, and while normally/abnormally breathing for different calibration methods.	
Witt et al., 2006 [64]	10 M Age (year): 23.4 ± 2.3 Mass (kg): 73.7 ± 7.8 Height (cm): 178.2 ± 8.6		Participants were monitored while standing rest.	Participants were monitored while performing different tasks, i.e., submaximal treadmill exercise, incremental treadmill test to exhaustion, recovery.

^1^ Age and anthropometrical features (mass, height, body mass index (BMI), and chest circumference (chest circumf.)) are reported as mean ± standard deviation and/or (range). Gender is referred to as female (F) and male (M). Acronyms: **CHF**: congestive heart failure; **COPD**: chronic obstructive pulmonary disease; **FGB**: fiber Bragg grating; **IR**: infrared; **OSA**: obstructive sleep apnea; **N/A**: not available; **RLD**: restrictive lung disease.

**Table 4 sensors-21-04124-t004:** Assessment of the accuracy.

Reference	Quasi-Static Motor Tasks		Dynamic Motor Tasks	
	Volumetric Error ^1^	Goodness of the Fitting Model ^2^	Volumetric Error ^1^	Goodness of the Fitting Model ^2^
Allsop et al., 2007 [34]	**RelErr** (1 SD): range [%] = 6–12	**ρ**: range = 0.2–0.9		
Angelucci et al., 2020 [35]		**R**^2^: 0.6707		
Antonelli et al., 2020 [31]	**RelErr** (median ± IQR): range [%] = (0.8 ± 33.2)–(9.0 ± 26.0)		**RelErr** (median ± IQR): range [%] = (−11.1 ± 29.4)–(−39.7 ± 13.8)	
Bates et al., 2010 [13]		**ρ**: value = 0.9597		
Brullman et al., 2010 [36]	**MOD** (mean ± SD): range [L] = (0.00 ± 0.04)–(0.04 ± 0.09)		**MOD** (mean ± SD): value [L] = 0.01 ± 0.12	
Caldiroli et al., 2007 [37]		**ρ**: range = 0.92–0.98		
Caretti et al., 1994 [38]	**RelErr** (mean ± SD): range [%] = (−0.35 ± 2.72)–(1.25 ± 3.68)		**RelErr** (mean ± SD): range [%] = (3.57 ± 2.70)–(11.52 ± 31.25)	**ρ** (mean ± SD): range = (0.89 ± 0.10)–(0.91 ± 0.06)
Chen et al., 2014 [39]	**MOD** (mean ± SD): value [L] = 0.091 ± 0.0704			
Chu et al., 2019 [6]	**MOD** (mean ± SD): value [L] = −0.077 ± 0.279	**ρ**: range = 0.929–0.962		
Clarenbach et al., 2005 [40]			**MOD** (mean ± SD): value [L] = −0.01 ± 0.32	
Coca et al., 2010 [41]			**MOD** (mean ± SD): value [L] = −0.03 ± 0.06	**ρ** (mean ± SD): value = (0.60 ± 0.12)
Cohen et al., 1997 [42]	**MOD** (mean ± SD): range [L] = (0.077 ± 0.072)–(0.255 ± 0.210)			
Earthrowl-Could et al., 2001 [43]	**MOD** (mean ± SD): range [L] ~ (−0.7 ± 0.35)–(5 ± 2)	**ρ**: range = 0.963–0.968		
Fekr et al., 2014 [44]		**ρ**: range = 0.77–0.91		
Gargiulo et al., 2015a [45]	**abs(RelErr)**: value [%] < 10%	**ρ**: value = 0.92		
Gargiulo et al., 2015b [46]	**abs(RelErr)**: value [%] ≤ 10%			
Grossman et al., 2010 [47]			**MOD** (mean ± SD): value [L] = 0.00 ± 0.102	**ρ** (mean ± SD): value = (0.89 ± 0.11)
Heyde et al., 2014 [48]	Inspiration **MOD** (mean ± SEE): range [L] = (−0.021 ± 0.106)–(0.041 ± 0.114) Exhalation **MOD** (mean ± SEE): range [L] = (−0.040 ± 0.115)–(0.047 ± 0.121)		Inspiration **MOD** (mean ± SEE): value [L] = −0.008 ± 0.088 Exhalation **MOD** (mean ± SEE): value [L] = −0.013 ± 0.097	
Heyde et al., 2015 [49]	**MOD** (mean ± SD): value [L] = 0.006 ± 0.069		**MOD** (mean ± SD): range [L] = (−0.007 ± 0.039)–(0.047 ± 0.063)	
Hoffman et al., 2011 [19]		**ρ** (mean ± SD): range = (0.443 ± 0.706)–(0.966 ± 0.032)		**ρ** (mean ± SD): range = (0.888 ± 0.085)–(0.940 ± 0.036)
Ivanovic et al., 2016 [50]	**abs(RelErr)** (mean ± SD): range [%] = (10.5 ± 3.8)–(15.0 ± 4.8)			
Kim et al., 2007 [51]		**ρ**: value = 0.96		
Laufer et al., 2021 [52]	**RMSE** (mean ± SD): value [L] = 0.155 ± 0.047	**ρ** (mean ± SD): value = 0.972 ± 0.020		
Làzaro et al., 2020 [53]		**ρ** (mean ± SD): value = 0.732 ± 0.066		
Lo Presti et al., 2018 [54]	**MOD** (mean ± SD): value [L] = 0.080 ± 0.191			
Massaroni et al., 2016 [55]	**MOD** (mean ± SD): value [L] = −0.058 ± 0.023			
Massaroni et al., 2018 [56]	**MOD** (mean ± SD): value [L] = −0.09 ± 0.15	**R**^2^: value = 0.90		
Padasdao et al., 2018 [57]	**abs(RelErr)** (mean ± SD): range [%] = (37.16 ± 13.74)–(42.37 ± 17.79) **MOD** (mean ± SD): range [L] = (0.001 ± 0.194)–(−0.008 ± 0.212)		**abs(RelErr)** (mean ± SD): value [%] = 31.15 ± 10.22 **MOD** (mean ± SD): value [L] = −0.021 ± 0.225	
Petrovic et al., 2014 [58]	**abs(RelErr)** (mean ± SD): range [%] = (10.5 ± 3.8)–(15.0 ± 4.8) **RelErr** (mean ± SD): range [%] = (−2.9 ± 11.5)–(−1.0 ± 9.9)			
Que et al., 2002 [59]	**abs(RelErr)** (mean ± SD): range [%] = (9.8 ± 3.1)–(18.7 ± 7.9) **MOD** (mean ± SD): range [L] = (−0.019 ± 0.061)–(0.015 ± 0.071)			
Raicevic et al., 2015 [60]	**abs(RelErr)**: value [%] = 10.8 ± 3.8			
Retory et al., 2016 [61]			**MOD** (mean ± SD): value [L] = −0.04 ± 0.24	**ρ**: value = 0.81
Seppa et al., 2013 [62]	**abs(RelErr)** (mean ± SD): range [%] = (6.7 ± 6.9)–(7.5 ± 2.0)			
Wang et al., 2020 [12]	**MOD** (mean ± SD): value [L] = 0.009 ± 0.190	**ρ** (mean ± SD): value = 0.89 ± 0.05		
Whyte et al., 1991 [63]	**MOD** (mean ± SD): range [L] = (−0.023 ± 0.054)–(0.026 ± 0.041)	**ρ** (mean ± SD): range = (0.56 ± 0.05)–(−0.65± 0.09)		
Witt et al., 2006 [64]	**abs(RelErr)**: value [%] = 14 ± 10 **MOD** (mean ± SD): value [L] = 0.13 ± 0.16		**abs(RelErr)** (mean ± SD): range [%] = (9 ± 11)–(15 ± 12) **MOD** (mean ± SD): range [L] = (0.13 ± 0.25)–(0.20 ± 0.39)	**R**^2^: value = 0.8743

^1^ Measures referring to the volumetric error adopted among studies are: mean of difference between estimated and measured tidal volume (MOD); relative error, i.e., ratio between MOD and estimated tidal volume expressed as percentage (RelErr), absolute relative error (abs(RelErr)), root-mean-squared error (RMSE); ^2^ measures referring to the goodness of the fitting model are the coefficient of determination (**R^2^**) and the correlation coefficient (**ρ**).Acronyms: **IQR**: interquartile range; **SEE**: standard error of the estimate; **SD**: standard deviation; **L**: liter.

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
