# Peer review of "Assessing the Tidal Volume through Wearables: A Scoping Review"

_sensors, 2021, doi:10.3390/s21124124_

Round 1
Reviewer 1 Report
This paper systematic summarized the use of wearable devices to monitor the tidal volume during both daily activities and clinical settings. It indicated that the relative volumetric error of these solutions (from IMUs to strain sensors or acoustic sensors) during quasi-static tasks typically is ≥10% but is deteriorates during dynamic motor tasks. Although, this paper told that additional efforts are required to improve the performance of these devices. If it can provide some suggestions and solutions methods about this limit should have more attractive to readers.
Reviewer 2 Report
The manuscript is a scoping review on the use of wearable/portable sensors for assessing tidal volume to get a complete picture of respiratory activity. The authors assessed wearable technology, especially in terms of accuracy, for measuring tidal volume. The review is meant to provide a guidance for future work in work with wearable for measuring respiratory activity. The theme seems quite time appropriate. This could have been a systematic review, but the authors have leaned towards a scoping review since this is not providing a concrete guidance, and some of the technology cited is still evolving. The authors wish to justify, to some extent, use of wearables for measuring tidal volume. In that regard, they could have discussed appropriateness of cited technology as a wearable.
- The authors could have added a few more appropriate keywords. While “tidal volume” and “wearable sensors” are fine, “assessment” and “accuracy” are general terms and not very appropriate for defining this work.
- Exclusion/inclusion criteria for article selection process
- It is unclear what the basis for adding the last five documents to the list was?
- The authors should have included Google Scholar while selecting articles. While may not be true for all cases, but a few articles on Google Scholar may not have be indexed elsewhere.
- The oldest of the articles are from the nineties. Technology and relevant research are continuously evolving. Did the authors consider a limit on the publishing year of the articles?
- While it is mentioned in the text, the flow chart on Figure 1 is missing the exclusion criteria.
- Authors excluded work which required a mask or a similar support mechanism (section 2.3, bullet point 2). Masks usually are not invasive unless you would need pipes to connect to a wearer’s mouth or nostrils. How complicated were those pieces of technology that were excluded from your analysis? Could the methods be modified or adapted to a wearable/portable platform?
The authors seemed to have some disinterest towards the consideration of smart spirometers as portables/wearables. They could reconsider as spirometry is minimally invasive, and may still work well with kids, especially portable smart spirometers that can fit in one’s pocket.
- The paper lists a good number of works, quite thorough, where the volumetric error for quasi-static tasks was greater than or equal to 10% while that for dynamic tasks was even higher. The review was not comparing apples-to-apples, and so as authors of the review paper, you may want to comment on how one approach or method could be the more appropriate way, and if and how selection and positioning of sensors improve overall results.
The authors could have included a stronger take-home message. Their discussion leading to the conclusion seemed less of a suggestion and more of an outline of limitations and possibilities. It is quite fine since this is a scoping review, but what would be your recommendation for current and future research in this area?
- All the tables have a heading in the first page only. Please included headings in all subsequent pages of each table for ease of understanding of the reader.
- The reference section has extra numbering. The authors may have submitted the article without taking a second look at that section.
- Please proofread for minor grammatical error and typo.
Reviewer 3 Report
The manuscript aims at identifying research evidence supporting the use of wearable devices to monitor the tidal volume during both daily activities and clinical settings. The content of review is not well organized, especially the section of references is riddled with errors.
Comment 1:
Part of the work on wearable sensors used "flexible" instead of "wearable" as the keyword, meanwhile part of the work on humidity sensor involved the test of "tidal volume". The expression needs to be unified throughout the article.
Comment 2:
Table 3 summarizes the information of subjects in the work investigated, but does not analyze the differences in test results caused by different subjects.
Comment 3:
The author should add comparative analysis of the measurement accuracy of sensors with different working modes.
Readers can not intuitively feel the differences between the sensors with different working modes, and it is difficult to judge which working mode has better performance.
Comment 4:
The author should offer suggestions on how to improve the performance of wearable devices used to monitor "tidal volume" in the future. The author can put forward some suggestions on the shortcomings of the existing work from the aspects of sensor structure design, sensing principle and testing method.
Comment 5:
The reference section should be thoroughly examined and corrected.
Reviewer 4 Report
In this paper the authors aim to assess of respiratory activity based on wearable devices by a screening of the literature (Pubmed, Scop….) run in December 2020 to collect the main studies.
The review approach seems to be developed more methodologically than scientifically.
Actually I think that how the authors have collected their references, even if this is an accurate work, will not be useful for the scientist dealing with this research activity.
The title should be changed in:
"Assessing the tidal volume through wearables through a methodological approach: a scoping review"
I invite them:
1) to skip from the abstract the sentence (line 3) "A screening of the literature (Pubmed, Scopus, and Web of Science) was carried out in December 2020 to collect studies", and the rewrite the subsequent text.
2) in the keyword change "accuracy" (line 25);
3) sub-paragraph 3.1 should be moved into paragraph 2.0 since it explains methodological details.
4) the references style should be revised.
Actually I guess that the authors aim to identify research evidences supporting the use of wearable devices to monitor the tidal volume in both daily activities and clinical settings, but perhaps they should not explain in details how they have collect the thematic bibliography on the wearables state of the art.
The work is well written, well organized, but they did not insert any explanatory images. I think that this paper could meet the basic requirements of Sensors, but major revision should be made before its acceptance for publication.
Round 2
Reviewer 3 Report
The authors have modified the content of sensors-1226961 following the reviewers' suggestions to meet the quality of Sensors, so I consider it to be published.
Reviewer 4 Report
The paper is suitable for publication in the present form.